# Investigating Different Context Types and Representations for Learning Word Embeddings

**Bofang Li, Tao Liu, & Zhe Zhao**
School of Information
Renmin University of China
Beijing, P.R. China
{libofang, tliu, helloworld}@ruc.edu.cn

**Buzhou Tang**
Department of Computer Science
Harbin Institute of Technology
Shenzhen, Guangdong, P.R. China
tangbuzhou@gmail.com

**Xiaoyong Du**
School of Information
Renmin University of China
Beijing, P.R. China
duyong@ruc.edu.cn

## Abstract

The number of word embedding models is growing every year. Most of them learn word embeddings based on the co-occurrence information of words and their context. However, it's still an open question what is the best definition of context. We provide the first systematical investigation of different context types and representations for learning word embeddings. We conduct comprehensive experiments to evaluate their effectiveness under 4 tasks (21 datasets), which give us some insights about context selection. We hope that this paper, along with the published code, can serve as a guideline of choosing context for our community.

## 1 Introduction

Recently, there is a growing research interest on word embedding models, where words are embedded into low-dimensional real vectors. Words that share similar meanings tend to have short distances in the vector space. The trained word embeddings are not only useful by themselves (e.g. used for calculating word similarities) but also effective when used as the input of the downstream models, such as chunking, tagging (Collobert & Weston, 2008; Collobert et al., 2011), parsing (Socher et al., 2011), text classification Socher et al. (2013); Kim (2014) and speech recognition (Schwenk, 2007).

For almost all word embedding models, the training objectives are based on Distributed Hypothesis (Harris, 1954), which can be stated as: "words that occur in the same contexts tend to have similar meanings". The "context" is usually defined as the words which precede and follow the target word within some fixed distance in most word embedding models with various architectures (Bengio et al., 2003; Mnih & Hinton, 2007; Mikolov et al., 2013b; Pennington et al., 2014). Among them, Global Vectors (GloVe) proposed by Pennington et al. (2014), Continuous Skip-Gram (CSG) [1] and Continuous Bag-Of-Words (CBOW) proposed by Mikolov et al. (2013a) achieve the state-of-the-art results on a wide range of linguistic tasks, and scales well to corpus with billion words.

Since the simplest way of defining context is used by these classic word embedding models, it is worth investigating the best definition of "context". For example, 1) the "context" can also be defined as the syntactic neighbours of the target word based on dependency parse tree. Is dependency-based context more reasonable than linear context defined by the positional neighbours of the target word in plain texts? 2) do the relative position of each contextual word and the relation between

---

[1] Many researches refer Continuous Skip-Gram as SG. However, in order to distinguish linear (continuous) context and dependency-based context, we refer it as CSG.

Table 1: Generalized Skip-Gram, Bag-Of-Words and GloVe with different context types and context representations. For linear context, *bound word* indicates word associated with positional information. For dependency-based context, *bound word* indicates word associated with dependency relation.

| basic model | context type / context representation | linear | dependency-based |
|---|---|---|---|
| generalized Skip-Gram | word | CSG (Mikolov et al., 2013a) | this work |
| | bound word | Structured SG (Ling et al., 2015) POSIT (Levy & Goldberg, 2014a) | Deps (Levy & Goldberg, 2014c) |
| generalized Bag-Of-Words | word | CBOW (Mikolov et al., 2013a) | this work |
| | bound word | CWINDOW (Ling et al., 2015) | this work |
| generalized GloVe | word | GloVe (Pennington et al., 2014) | this work |
| | bound word | this work | this work |

contextual word and target word contribute to the learning process? 3) do different word embedding models have preference for different context? This paper tries to answer these questions based on the experimental results according to different tasks.

Previously, Levy & Goldberg (2014a); Ling et al. (2015) [2] improve the CSG and CBOW by introducing position-aware context, where each contextual word is associated with their relative position to the target word. Levy & Goldberg (2014c) proposes DEPS, which considers the words that are connected to target word in dependency parse tree as context. We classify these models based on different context types (linear or dependency-based) and different context representations (word or bound word) in Table 1. We implement the models that previously not proposed and give systematical comparisons of different context types and context representations on popular CSG, CBOW, and GloVe. Comprehensive experiments are conducted on a wide range of word similarity, word analogy, sequence labeling, and text classification datasets. Some insights about determining the context in different situations are presented. We expect this paper to be an useful complementary in the word embedding literature.

## 2 METHODOLOGY

### 2.1 CONTEXT TYPES

It is necessary to discover more effective ways to define "context". In the current literature, there are two types of context: linear (most word embedding models) and dependency-based (DEPS (Levy & Goldberg, 2014c)). Linear context is defined as the positional neighbours of the target word in text. Dependency based context is defined as the syntactic neighbours of the target word based on dependency parse tree, as shown in Figure 1 [3].

Compared to linear context, dependency-based context can capture more long-range context. For example, linear context does not consider the word-context pair (discovers, telescope), while dependency-based context contains these information. Dependency-based context can also exclude some uninformative word-context pairs like (with, star) and (telescope, with).

### 2.2 CONTEXT REPRESENTATIONS

In the CSG and CBOW, context is represented by words without additional information. Levy & Goldberg (2014a); Ling et al. (2015) improve them by introducing position-bound words, where each contextual word is associated with their relative position to the target word. This allows CSG

---

[2]In these two papers, the description of position-aware context are quite different. However, their ideas is actually identical.

[3]This example is originally shown in Levy & Goldberg (2014c)

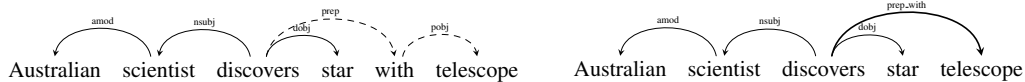

Figure 1: Illustration of dependency parse tree for sentence "Australian scientist discovers star with telescope". Note that preposition relation is collapsed in the right sub-figure, where *telescope* is considered as a direct modifier of *discovers*.

and CBOW to distinguish different sequential positions and capture context's structural information. We name the method that bind additional information to contextual word as bound context representation, as opposite to unbound context representation where word is used alone.

For dependency-based context, the original DEPS uses bound context representation by default: words are associated with their dependency relation to the target word. Similar to bound context representation in linear context type, this allows word embedding models to capture more dependency information. An example is shown in Table 2

Table 2: Illustration of bound and unbound context representations under linear and dependency-based context types. This example is based on Figure 1 and the target word is "discovers".

| context type / context representation | linear | dependency-based |
|---|---|---|
| unbound | australian, scientist, star, with | scientist, star, telescope |
| bound | australian/-2, scientist/-1, star/+1, with/+2 | scientist/nsubj, star/dobj, telescope/prep_with |

Note that bound context representation is sparse, especially for dependency-based context. There are 47 dependency relations in dependency parse tree. Although not every combination of dependency relations and words appear in the word-context pair collection, it still enlarges the context vocabulary about 5 times in practice. In this paper, we investigate the simpler context representation where no dependency relation are considered. This also makes a fair comparison with linear context models like CSG, CBOW and GloVe, since they do not use bound context representation neither.

Table 3: Illustration of collection $P$, $M$ and $\overline{M}$ for sentence "australian scientist discovers star with telescope". Unbound context representation is used in this example. Words in the collections are **Bold** and contexts in the collections are Normal.

| | linear (window size equals 1) | dependency-based |
|---|---|---|
| $P$ | (**australian**, scientist)<br>(**scientist**, australian) (**scientist**, discovers)<br>(**discovers**, scientist) (**discovers**, star)<br>... | (**australian**, scientist)<br>(**scientist**, australian) (**scientist**, discovers)<br>(**discovers**, scientist) (**discovers**, star)<br>(**discovers**, telescope)<br>... |
| $M$ | (**australian**, scientist)<br>(**scientist**, australian, discovers)<br>(**discovers**, scientist, star)<br>... | (**australian**, scientist)<br>(**scientist**, australian, discovers)<br>(**discovers**, scientist, star, telescope)<br>... |
| $\overline{M}$ | (**australian**, scientist, 1)<br>(**scientist**, australian, 1) (**scientist**, discovers, 1)<br>(**discovers**, scientist, 1) (**discovers**, star, 1)<br>... | (**australian**, scientist, 1)<br>(**scientist**, australian, 1) (**scientist**, discovers, 1)<br>(**discovers**, scientist, 1) (**discovers**, star, 1)<br>(**discovers**, telescope, 1)<br>... |

## 2.3 GENERALIZATION

For convenient and general representation, we first define the collection of word-context pairs as $P$. $P$ can be merged based on the words to form a collection $M$ with size of $|C|$. Each element $(w, c_1, c_2, .., c_{n_w}) \in M$ is the word $w$ and its contexts, where $n_w$ is the number of word $w$'s contexts. $P$ can also be merge based on both words and contexts to form a collection $\overline{M}$. Each element $(w, c, \#(w, c)) \in \overline{M}$ is the word $w$, context $c$, and the times they appears in collection $P$. An example of these collections is shown in Table 3.

### 2.3.1 GENERALIZED BAG-OF-WORDS

The objective function of Generalized Bag-Of-Words (GBOW) with negative sampling technique is defined as:

$$\sum_{(w, c_1, .., c_{n_w}) \in M} \log p \left( w \middle| \sum_{i=1}^{n_w} \vec{c_i} \right) = \sum_{(w, c_1, .., c_{n_w}) \in M} \left[ \log \sigma \left( \vec{w} \cdot \sum_{i=1}^{n_w} \vec{c_i} \right) - \sum_{k=1}^{K} \log \sigma \left( \vec{w_N} \cdot \sum_{1=i}^{n_w} \vec{c_i} \right) \right] \tag{1}$$

where $\sigma$ is the sigmoid function, $K$ is the negative sampling size, $\vec{w}$ and $\vec{c}$ is the vector for word $w$ and $c$ respectively. The negatively sampled random word $w_N$ is selected based on its unigram distribution $(\frac{\#(w)}{\sum_w \#(w)})^{ds}$, where $\#(w)$ is the number of times that word $w$ appears in the corpus, $ds$ is the distribution smoothing hyper-parameter which is usually defined as $0.75$.

Note that in original CBOW with negative sampling technique, the probability is actually $p\left(c| \sum \vec{w_i}\right)$ instead of $p\left(w| \sum \vec{c_i}\right)$. In other word, original CBOW uses the sum of word vectors to predict context. This works well for linear context. But for dependency-based context with bound word, there is only one contextual word available for prediction. For example in Figure 1, the context "scientist/nsubj" can only be predicted by word "discovers". However, a word can be predicted by the sum of several contexts. Due to this reason, we exchange the role of word and context in GBOW. The negative sampling objective is also changed from context $c_N$ to word $w_N$.

### 2.3.2 GENERALIZED SKIP-GRAM

For generalized Skip-Gram (GSG), the definition is straightforward and actually need no modification of the objective function, as discussed in (Levy & Goldberg, 2014a). However, in order to make it consistent with our GBOW, we also exchange the role of word and context. the objective function of GSG is defined as:

$$\sum_{(w, c) \in P} \log p\left(w|\vec{c}\right) = \sum_{(w, c) \in P} \left[ \log \sigma\left(\vec{w} \cdot \vec{c}\right) - \sum_{k=1}^{K} \log \sigma\left(\vec{w_N} \cdot \vec{c}\right) \right] \tag{2}$$

### 2.3.3 GLOVE

Unlike GSG and GBOW, GloVe explicitly optimizes a log-bilinear regression model based on word co-occurrence matrix. Since GloVe is already a very generalized model, with our previous defined collection $\overline{M}$, the final objective function is easily written as:

$$\sum_{(w, c) \in \overline{M}} f(\#(w, c))(\vec{w} \cdot \vec{c} + \vec{b_w} + \vec{b_c} - \log \#(w, c)) \tag{3}$$

where $f$ is a non-decreasing weighting function and ensures the weight of large $\#(w, c)$ to be relatively small.

Note that the inputs of GSG, GBOW and Glove are the collection $P$, $M$ and $\overline{M}$ respectively. Once the corpus and hyper-parameters are fixed, these collections (and thus the learned word embedings) are determined only by the choice of context types and representations.

## 3 EXPERIMENTS

We evaluate the effectiveness of different context types and representations on word similarity task, word analogy task, sequence labeling task, and text classification task. In this Section, we first

describe the training details of word embedding models. We then report and discuss the experimental results on each task. The full experimental results can be found in the Appendix.

## 3.1 TRAINING DETAILS

The `word2vecf` toolkit [4] (Levy et al., 2015) extends the `word2vec` toolkit [5] (Mikolov et al., 2013b) to accept the input of collection $P$ rather than raw corpus. This makes CSG model accept any arbitrary contexts (e.g. dependency-based context). However, CBOW model is not considered in that toolkit. We implement `word2vecPM` [6], a further extension of `word2vecf`, which supports both generalized SG and generalized BOW with the input of collection $P$ and $M$ respectively.

We use English Wikipedia (August 2013 dump) as the training corpus in all of our experiments. The Stanford CoreNLP (Manning et al., 2014) is used for dependency parsing. All words and contexts are converted to lower case after parsing. Words and contexts that appear less than 100 times in collection $P$ and $M$ are directly ignored. Note that this is slightly different from ignoring rare word that appear less than 100 times in corpus, since each word may appear more times in collection than that in corpus.

Most hyper-parameters are the same as Levy et al. (2015)'s best configuration. For example, negative sampling size $K$ is set to 5 for GSG and 2 for GBOW. Distribution smoothing $cds$ is set to 0.75. No dynamic context or "dirty" sub-sampling is used. The window size $wn$ is fixed to 2 for constructing linear context, which insures the number of the (merged) word-context pair collection for both linear context and dependency-based context is comparable. The number of iteration is set to 2, 5 and 30 for GSG, GBOW and GloVe respectively. Unless otherwise noted, the number of word embedding dimension is set to 500. Since the aim of this paper is not comparing the performance of different word embedding models, the results of GSG, GBOW and GloVe are reported respectively.

## 3.2 WORD SIMILARITY TASK

Word similarity task aims at producing a semantic similarity score of a word pair, which is compared with the human label. The cosine distance is used for scoring similarities between two words, and measured by Spearman's correlation. Six datasets are used in our experiments: WordSim353 (Finkelstein et al., 2001) with similarity and relatedness partition (Zesch et al., 2008; Agirre et al., 2009), MEN dataset (Bruni et al., 2012), Mechanical Turk dataset (Radinsky et al., 2011), Rare Words dataset (Luong et al., 2013), SimLex-999 dataset (Hill et al., 2016).

Table 4: Results on 6 word similarity datasets. Best results in group are marked **Bold**.

| model | context type | context rep | WS353 sim | WS353 related | MEN | Mech Turk | Rare Words | SimLex 999 |
|---|---|---|---|---|---|---|---|---|
| GSG | linear | word | .757 | **.563** | **.732** | .632 | .414 | .417 |
| | | bound | .762 | .543 | .695 | .608 | .421 | **.434** |
| | dep | word | .776 | .531 | .728 | **.644** | **.422** | .418 |
| | | bound | **.792** | .483 | .674 | .643 | .413 | .421 |
| GBOW | linear | word | .747 | **.503** | **.718** | **.644** | **.436** | **.439** |
| | | bound | .689 | .427 | .659 | .512 | .403 | .428 |
| | dep | word | .669 | .395 | .667 | .541 | .412 | .386 |
| | | bound | **.799** | **.502** | .640 | .587 | **.434** | .403 |
| GloVe | linear | word | .645 | **.545** | .662 | .587 | .354 | .323 |
| | | bound | .670 | .481 | .563 | .587 | .400 | .363 |
| | dep | word | .696 | .539 | **.692** | **.603** | .371 | .342 |
| | | bound | **.734** | .468 | .541 | .557 | **.409** | **.406** |

As shown in the numerical results in Table 4, there is no single model consistently outperform the rest across all datasets. Although the overall trend of GSG, GBOW and GloVe using different context types and representations is similar, GBOW seems more benefit from linear context than

---

[4] https://bitbucket.org/yoavgo/word2vecf
[5] http://code.google.com/p/word2vec/
[6] https://github.com/libofang/word2vecPM

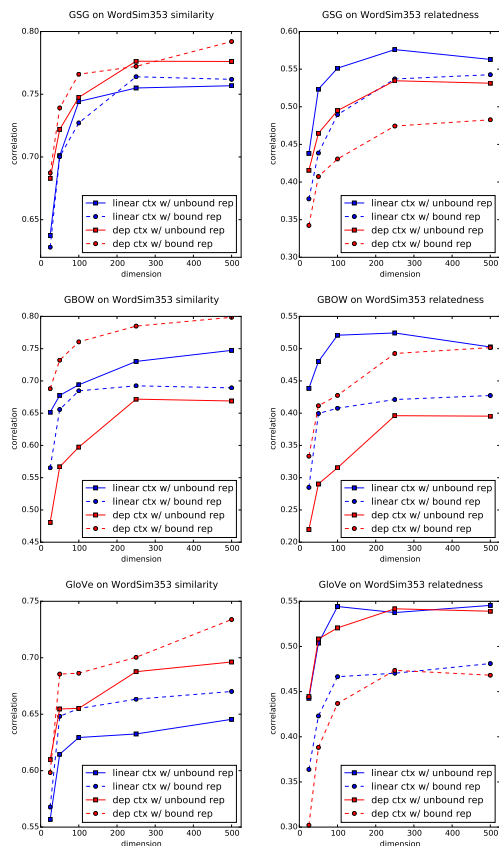

Figure 2: Results on WordSim353 (similarity and relatedness) datasets.

GSG and GloVe. GBOW takes the sum of context vectors as prediction function's input, thus is less sensitive syntactic structure. In other words, since the "right" context is summed with other context, it's information contributes less than that in GSG.

More conclusion could be conducted if we focus on the WordSim353 dataset with similarity and relatedness partition. It's previously commonly believed that compared to linear context, dependency-based context can capture more functional similarity (e.g. tiger/cat) rather than topical similarity/relatedness (e.g., tiger/jungle) (Levy & Goldberg, 2014c; Melamud et al., 2016). However, these experiments do not distinguish the effect of different context representations: unbound representation is used for linear context ((Mikolov et al., 2013b)) while bound representation is used for dependency-based context ((Levy & Goldberg, 2014c)). Moreover, only CSG model is compared.

We revisit previous claims based on more systematical results. As shown in Figure 2's upper-left sub-figure, compared to linear context (solid and dotted blue line), the better results of dependency-based context for GSG and GloVe (solid and dotted red line) on ws353's similarity partition confirms its ability of capturing functional similarity. However, the good performance of dependency-based context for GSG do not fully transfer to GBOW. Although dependency-based context with bound representation for GBOW is still the best performer, dependency-based context with unbound representation for GBOW (solid red line) performs worst on ws353's similarity partition. Note that the results are also reversed on ws353's relatedness partition (Figure 2's right sub-figures), which shows the use of linear context is more suitable for capturing topical relatedness.

### 3.3 WORD ANALOGY TASK

Word analogy task aims at answering the question like "a is to b as c is to __ ?". For example, "London is to UK as Tokyo is to Japan". We follow the evaluation protocol in Levy & Goldberg

Table 5: Results on 6 word similarity datasets and 3 word analogy datasets. Best results in group are marked **Bold**.

| model | context type | context rep | Google Sem | Google Syn | MSR | Inflectional morphology | Derivational morphology | Encyclopedic | Lexicographic |
|---|---|---|---|---|---|---|---|---|---|
| GSG | linear | word | .708 | .639 | .642 | .678 | .110 | .242 | .083 |
| | | bound | .702 | .454 | **.653** | .668 | .111 | .208 | **.099** |
| | dep | word | **.716** | **.661** | .644 | **.691** | **.122** | **.253** | .095 |
| | | bound | .600 | .307 | .600 | .668 | .112 | .170 | **.099** |
| GBOW | linear | word | **.628** | **.566** | **.601** | **.618** | **.096** | .201 | .074 |
| | | bound | .602 | .376 | .569 | .572 | .091 | .157 | **.081** |
| | dep | word | .573 | .553 | .520 | .496 | .094 | **.216** | .076 |
| | | bound | .495 | .248 | .516 | .563 | .086 | .126 | .078 |
| GloVe | linear | word | .471 | **.719** | .454 | .425 | .033 | .226 | .054 |
| | | bound | .502 | .218 | **.542** | **.559** | **.044** | .129 | **.095** |
| | dep | word | **.513** | .700 | .525 | .491 | .043 | **.227** | .063 |
| | | bound | .402 | .121 | .525 | .446 | .033 | .093 | .083 |

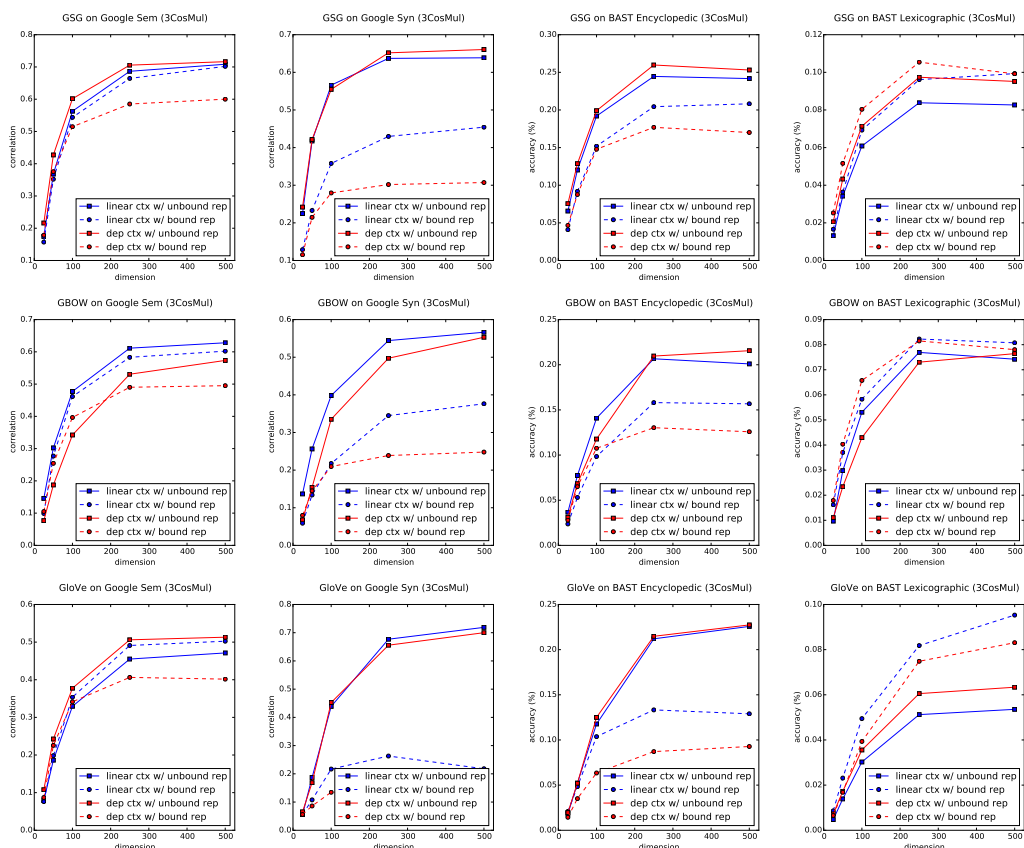

Figure 3: Results on Google (Sem and Syn) and BATS datasets (Encyclopedic and Lexicographic).

(2014a), answering the questions using both 3CosAdd and 3CosMul. Our experiments show that 3CosMul works consistently better than 3CosAdd, thus only the results of 3CosMul are reported. We follow previous researches, use Google's analogy dataset (Mikolov et al., 2013a) (with semantic and syntactic partition), MSR's analogy dataset (Mikolov et al., 2013c), and BATS analogy dataset (Gladkova et al., 2016) in our experiments.

Numerical results are shown in Table 5. We observe that the context representation plays an important role in word analogy task. The choice of context representation (word or bound word) actually has much larger impact than the choice of context type (linear or dependency). The results on Google Syn dataset (Figure 3's sub-figures in second column) is perhaps most evident. The performance of linear context and dependency-based context with unbound representation is similar.

However, when bound context representation is used, the performance of GSG and GBOW drops more than 30 percent for dependency-based context and around 20 percent for linear context. The main reason for this phenomenon may be that the bound representation already contains syntactic information, thus word embedding models can not learn it from the input word-context pairs. It can also be observed that GloVe is more sensitive to different context representations than Skip-Gram and CBOW, which is probably due to its explicitly defined/optimized objective function.

## 3.4 SEQUENCE LABELING TASK

Although intrinsic evaluations like word similarity and word analogy tasks could provide direct insights of different context types and representations, the experimental results above cannot be translated to typical uses of word embeddings. For example, these tasks aren't necessarily correlated with downstream tasks' accuracy, as shown in (Schnabel et al., 2015; Linzen, 2016; Chiu et al., 2016). More extrinsic tasks should be considered.

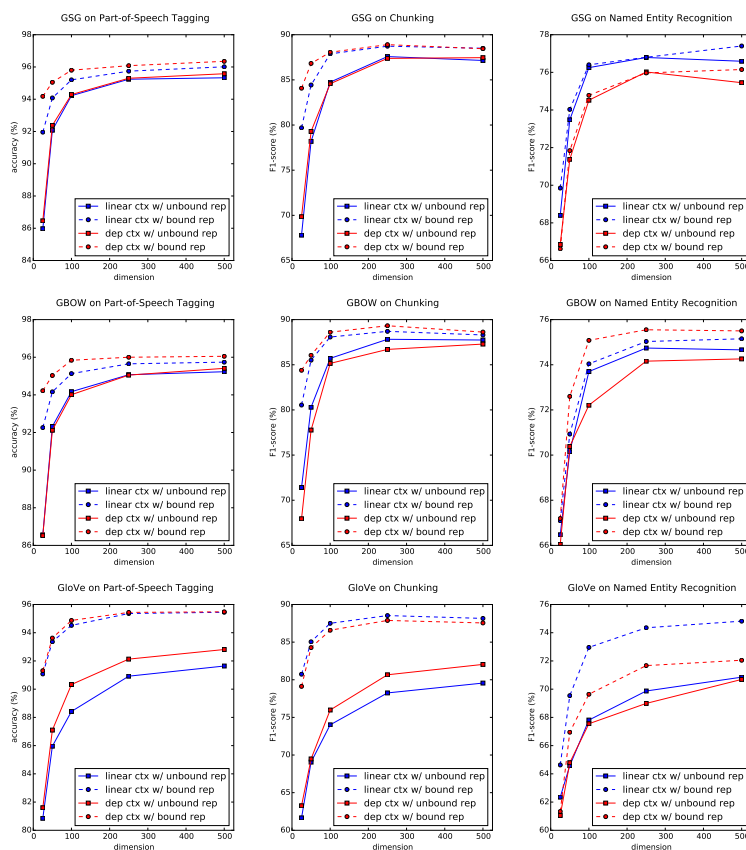

Figure 4: Results on sequence labeling task.

In this Subsection, we evaluate the effectiveness of different word embedding models with different contexts on sequence labeling task. Sequence labeling aims at automatically assigning words in texts with labels. Three sub-tasks are considered: Part-of-Speech Tagging (POS), Chunking and Named Entity Recognition (NER) . CoNLL 2000 shared task [7] is used as benchmark for POS and Chunking. CoNLL 2003 shared task [8] is used as benchmark for NER.

Recent advances on sequence labeling task are based on neural networks like Recurrent Neural Network, Convolutional Neural Network, and their combinations with Conditional Random Fields (Collobert et al., 2011; Huang et al., 2015; Ma & Hovy, 2016). These models all require word

---

[7]`http://www.cnts.ua.ac.be/conll2000/chunking`
[8]`http://www.cnts.ua.ac.be/conll2003/ner`

embeddings as input. Inspired by the evaluation protocol used in Kiros et al. (2015), we restrict the prediction to simple linear classifier. More precisely, the classifier's input for predicting the label of word $w_i$ is simply the concatenation of vector $\vec{w_{i-2}}$, $\vec{w_{i-1}}$, $\vec{w_i}$, $\vec{w_{i+1}}$, $\vec{w_{i+2}}$. This ensures the quality of embedding models is directly evaluated, and their strengths and weaknesses are easily observed.

As shown in Figure 4, the overall trend of GSG, GBOW and GloVe is identical except on NER task. Linear context type (red line) works better than dependency-based (blue line) context type when unbound context representation is used. The results are reversed when bound context representation is used. Bound context representation (dotted linear) outperforms unbound context representation (solid linear) on all datasets. These results suggest that linear context type with unbound context representations (as in traditional CSG and CBOW) may not be the best choice of input word vectors for sequence labeling. Dependency-based context with bound context representations should be used instead. Again, similar to that on word analogy task, GloVe is more sensitive to different context representations than Skip-Gram and CBOW on sequence labeling task.

## 3.5 TEXT CLASSIFICATION TASK

Finally, we evaluate the effectiveness of different word embedding models with different contexts on text classification task. Text classification is one of the most popular and well-studied task in natural language processing. Recently, deep neural networks are dominant on this task (Socher et al., 2013; Kim, 2014; Dai & Le, 2015). They often need pre-trained word embeddings as inputs to improve their performances. Similar to our evaluation of sequence labeling, instead of building complex deep neural networks, we use a simpler classification method called Neural Bag-of-Words to directly evaluate the word embeddings: texts are first represented by the sum of their belonging words' vectors, then a Logistic Regression Classifier is built upon them for classification.

Different word embedding models are evaluated on 5 text classification datasets. The first 3 datasets are sentence-level: short movie review sentiment (MR) (Pang & Lee, 2005), customer product reviews (CR) (Nakagawa et al., 2010), and subjectivity/objectivity classification (SUBJ) (Pang & Lee, 2004). The other 2 datasets are document-level with multiple sentences: full-length movie review (RT-2k) (Pang & Lee, 2004), and IMDB movie review (IMDB) (Maas et al., 2011).

As shown in Table 6, pre-trained word embeddings outperform random word embeddings by a large margin. This further strengthen previous researches that pre-trained word embeddings are crucial for text classification. Unlike that on previous tasks, different models' results are actually very similar on text classification task. Overall, models which use bound context representation perform worse than those which use unbound context representation on all datasets except CR. The performances of models that use dependency-based context type and linear context type is comparable. These observations suggest that simple linear context type with unbound context representations (as in traditional CSG and CBOW) is still the best choice of pre-training word embeddings, which is already used in most researches.

Table 6: Results on 5 text classification datasets.

| model | context type | context rep | sentence-level | | | document-level | |
|---|---|---|---|---|---|---|---|
| | | | MR | CR | Subj | RT-2k | IMDB |
| GSG | linear | word | 76.1 | 78.3 | 90.9 | 83.5 | 85.2 |
| | | bound | 75.3 | 79.0 | 90.4 | 82.2 | 85.2 |
| | dep | word | 76.0 | 77.7 | 90.7 | 84.8 | 85.1 |
| | | bound | 75.0 | 77.5 | 90.0 | 84.7 | 84.5 |
| GBOW | linear | word | 74.9 | 77.9 | 90.4 | 82.0 | 85.0 |
| | | bound | 74.1 | 77.8 | 90.3 | 80.7 | 84.1 |
| | dep | word | 75.0 | 77.6 | 90.1 | 82.4 | 84.9 |
| | | bound | 73.5 | 78.2 | 89.9 | 80.7 | 83.4 |
| GloVe | linear | word | 73.4 | 76.7 | 89.6 | 79.2 | 83.5 |
| | | bound | 73.2 | 77.5 | 90.0 | 79.8 | 83.4 |
| | dep | word | 74.0 | 77.7 | 89.5 | 81.3 | 83.5 |
| | | bound | 72.5 | 76.7 | 88.8 | 79.2 | 83.5 |
| random word embeddings | | | 63.9 | 72.8 | 79.9 | 72.2 | 77.2 |

# 4 RELATED WORK

Previously, there are researches which directly compare different word embedding models. **?** compares 6 word embedding models using different corpora and hyper-parameters. Levy & Goldberg (2014b) shows the theoretical equivalence of CSG and PPMI matrix factorization. Levy et al. (2015) further discusses the connections between 4 word embedding models (PPMI, PPMI+SVD, CSG, GloVe) and re-evaluates them with the same hyper-parameters. Suzuki & Nagata (2015) investigates different configurations of CSG and Glove, then merges them into a unified form. Yin & Schutze (2016) proposes 4 ensemble methods and shows their effectiveness over individual word embeddings.

There are also researches which focus on evaluating different context types in learning word embeddings. Vulic & Korhonen (2016) compares CSG and dependency-based models on various languages. The results suggest that dependency-based models are able to detect functional similarity on English. However, the advantages of dependency-based context over linear context on other languages is not as promising as that on English. Bansal et al. (2014) investigates different embedding models for parsing task and shows that dependency-based context is more suitable than linear context. Melamud et al. (2016) investigate the performance of CSG, Deps and a substitute-based word embedding models (Yatbaz et al., 2012) [9], which shows that different types of intrinsic task have clear preference to particular types of contexts. On the other hand, for extrinsic task, the optimal context types need to be carefully tuned on specific dataset. However, context representations (word and bound) are not evaluated in these models. Moreover, they focus only on CSG model since it's more general and intuitive for dependency-based context.

# 5 CONCLUSION

To the best of our knowledge, this paper provides the first systematical investigation of different context types and representations for learning word embeddings. We evaluate different models on 4 tasks with totally 21 datasets. Experimental results show that:

- Dependency-based context type does not get all the credit for capturing functional similarity. Bound context representation also plays an important role, especially for GBOW.
- Syntactic word analogy benefits less from bound context representation. Bound context representation already contains syntactic information, which makes it difficult to capture this information based on the input word-context pairs.
- Bound context representation is suitable for sequence labeling task, especially when it is used along with dependency-based context.
- On text classification task, different contexts do not affect the final performance much. Nonetheless, the use of pre-trained word embeddings is crucial and linear context type with unbound representation (Skip-Gram) is still the best choice.
- The overall tendency of models with different contexts is similar, especially for Skip-Gram and GloVe. GloVe is more sensitive to different contexts than Skip-Gram and CBOW. CBOW benefits most from linear context type.

In the spirit of transparent and reproducible experiments, the source code is published at `https://github.com/libofang/word2vecPM`. We hope researchers will take advantage of our code for further improvements and applications to other tasks.

ACKNOWLEDGMENTS

We thank Omer Levy, Yoav Goldberg, and Ido Dagan for their implementation of `word2vecf` and evaluation scripts. It systematically investigated the effective of different hyper-parameters on various word embedding models. Both their code and paper have directly influenced us a lot.

This work is supported by National Natural Science Foundation of China (Grant No. 61472428 and No. 71271211), the Fundamental Research Funds for the Central Universities, the Research Funds

---

[9]We do not consider this context type in this paper since it performs consistently worse than other two context types, as shown in Melamud et al. (2016); Vulic & Korhonen (2016)

of Renmin University of China No. 14XNLQ06. This work is partially supported by ECNU-RUC-InfoSys Joint Data Science Lab and a gift from Tencent.

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

APPENDIX

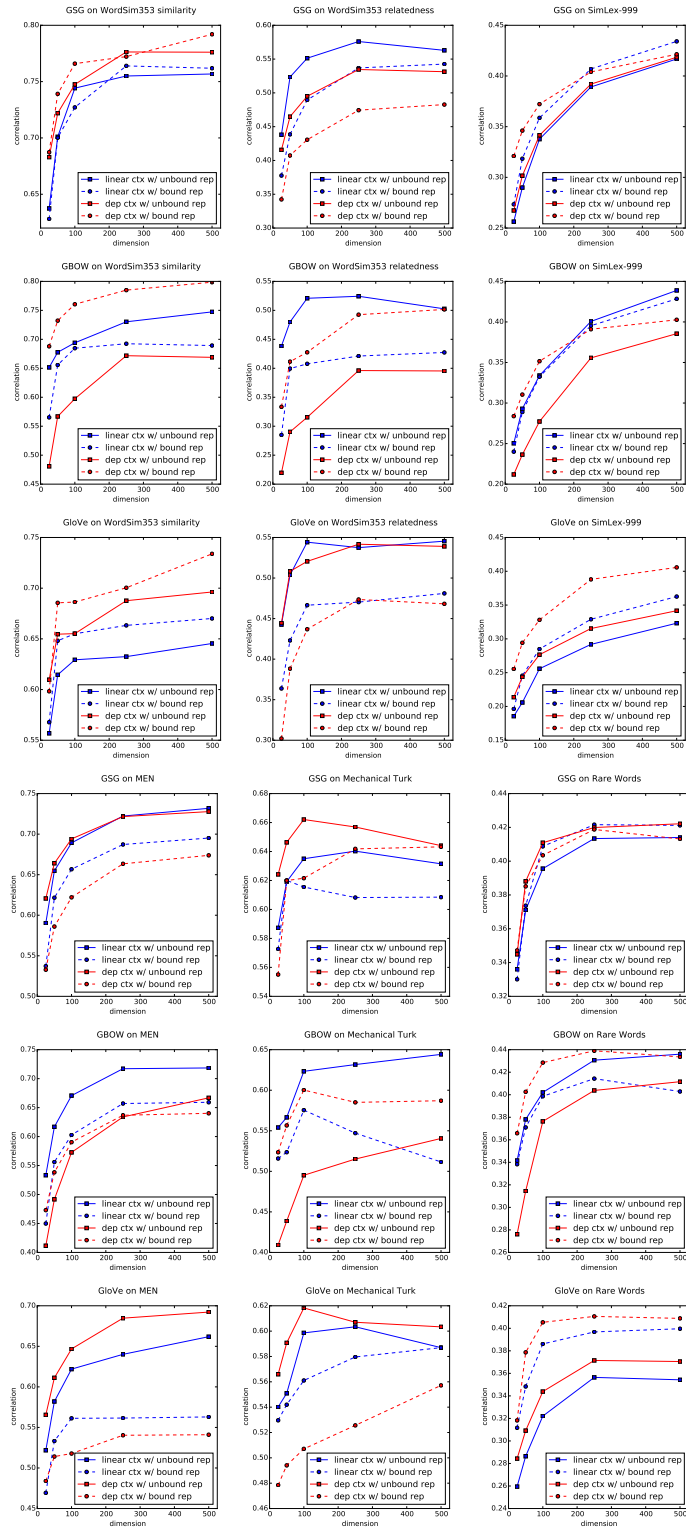

Figure 5: Full results on word similarity task.

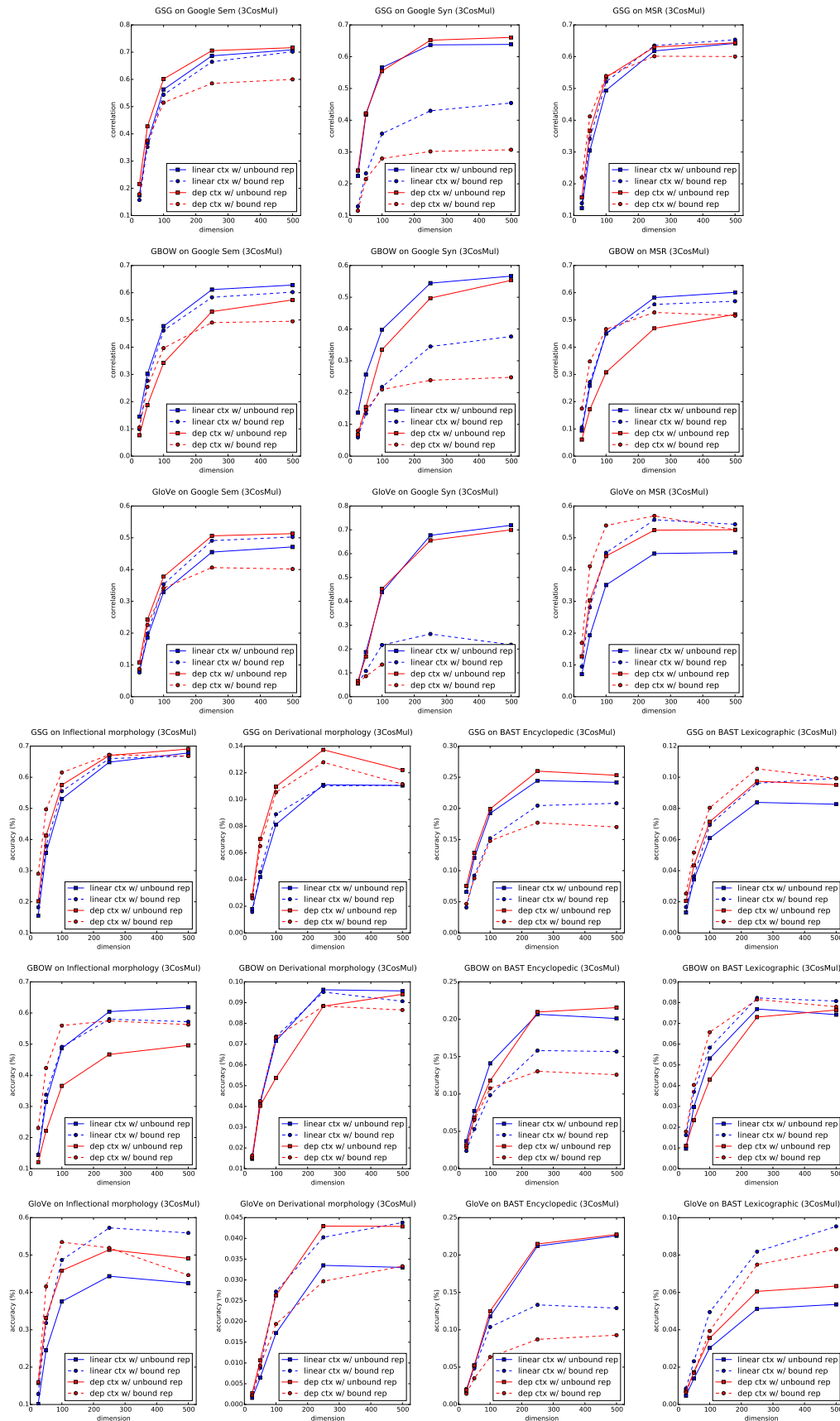

Figure 6: Full results on word analogy task.

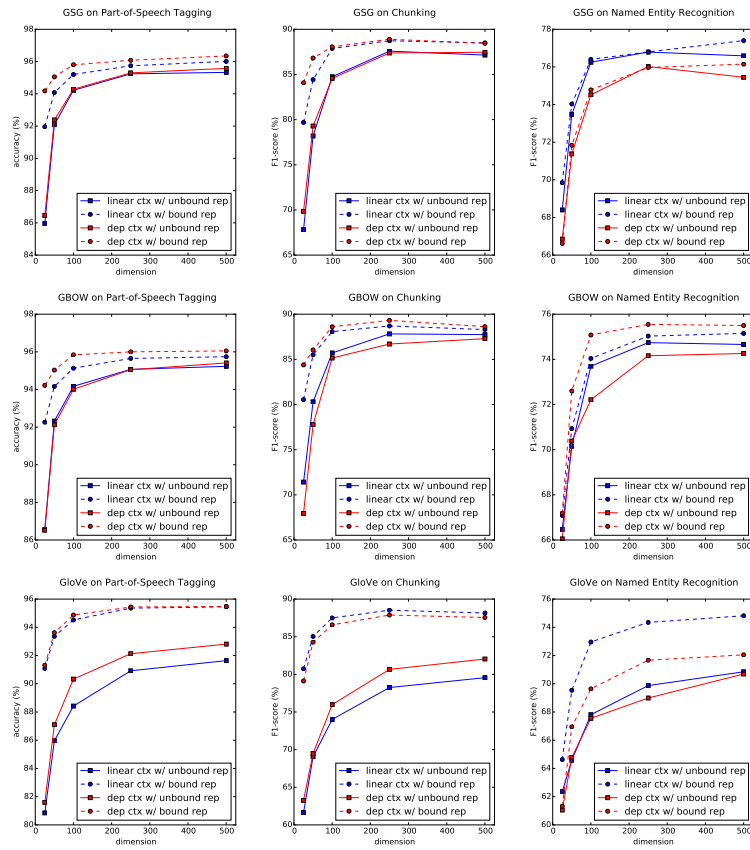

Figure 7: Full results on sequence labeling task.

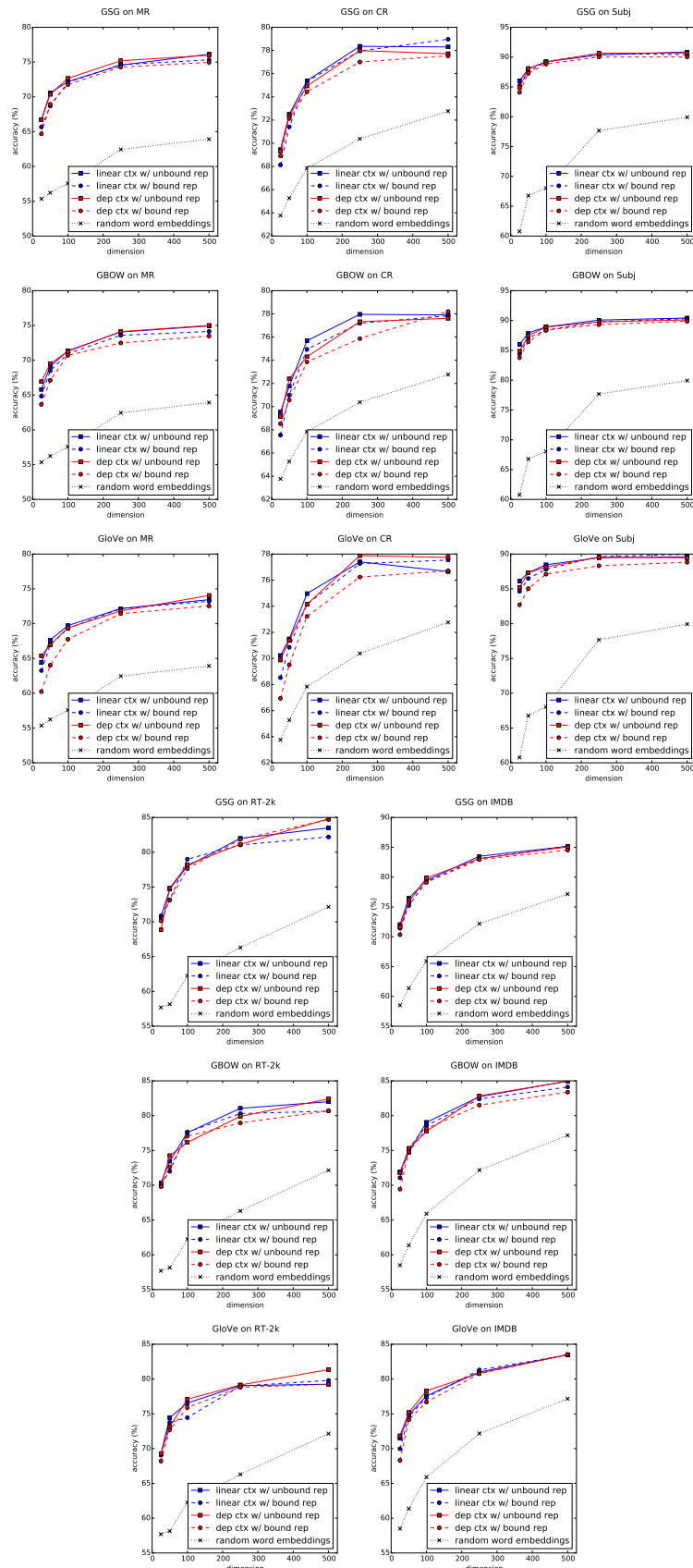

Figure 8: Full results on text classification task.

