# Peer review of "Investigating Different Context Types and Representations for Learning Word Embeddings"

_ICLR 2017 — rejected_

[Reviewer Comment · AnonReviewer1 · 02 Dec 2016]
**Additional Context Types**

There are many other types of contexts which should be discussed; see "Open IE as an Intermediate Structure for Semantic Tasks" (Stanovsky et al., ACL 2015).

[Official Review · AnonReviewer3 · rating 4 · confidence 3 · 11 Dec 2016]
**Solid work, but inconclusive and of narrow interest**

This paper investigates the issue of whether and how to use syntactic dependencies in unsupervised word representation learning models like CBOW or Skip-Gram, with a focus one the issue of bound (word+dependency type, 'She-nsubj') vs. unbound (word alone, 'She') representations for context at training time. The empirical results are extremely mixed, and no specific novel method consistently outperforms existing methods.

The paper is systematic and I have no major concerns about its soundness. However, I don't think that this paper is of broad interest to the ICLR community. The paper is focused on a fairly narrow detail of representation learning that is entirely specific to NLP, and its results are primarily negative. A short paper at an ACL conference would be a more reasonable target.

[Official Review · AnonReviewer1 · rating 6 · confidence 4 · 14 Dec 2016]
**No Title**

This paper evaluates how different context types affect the quality of word embeddings on a plethora of benchmarks.

I am ambivalent about this paper. On one hand, it continues an important line of work in decoupling various parameters from the embedding algorithms (this time focusing on context); on the other hand, I am not sure I understand what the conclusion from these experiments is. There does not appear to be a significant and consistent advantage to any one context type. Why is this? Are the benchmarks sensitive enough to detect these differences, if they exist?

While I am OK with this paper being accepted, I would rather see a more elaborate version of it, which tries to answer these more fundamental questions.

[Official Review · AnonReviewer2 · rating 4 · confidence 5 · 24 Dec 2016]
**Belowline**

This paper analyzes dependency trees vs standard window contexts for word vector learning.
While that's a good goal I believe the paper falls short of a thorough analysis of the subject matter.
It does not analyze Glove like objective functions which often work better than the algorithms used here.
It doesn't compare in absolute terms to other published vectors or models.

It fails to gain any particularly interesting insights that will modify other people's work.
It fails to push the state of the art or make available new resources for people.

[Public Comment · li bofang · 08 Jan 2017]
**revised version and response to all reviews**

Dear reviewers,

	We have added three sequence labeling tasks (POS, Chunking, and NER) and a word analogy dataset. 
	The models are currently evaluated on 4 tasks with 21 datasets. It is indeed hard to find any universal insight. However, after revisiting our experimental results and re-organizing the experiment section, we draw the following conclusions according to specific tasks and are excited to share with you:
		1.Dependency-based context type does not get all the credit for capturing functional similarity. Bound context representation also plays an important role, especially for GBOW.
		2.Syntactic word analogy benefits less from bound context representation. Bound context representation already contains syntactic information, which makes it difficult to capture this information based on the input word-context pairs.
		3.Bound context representation is suitable for sequence labeling task, especially when it is used along with dependency-based context.
		4.On text classification task, different contexts do not affect the final performance much. Nonetheless, the use of pre-trained word embeddings is crucial and linear context type with unbound representation is still the best choice.
	For more details, please see our revised experiment section.
	As for the GloVe model, we just need a few more days to re-run it for the best configuration. We promise that the results will be added before January 14th (Saturday). So sorry for the delay.
	We sincerely thank you for the detailed and constructive reviews. Please do kindly leave comments and suggests to help us further improve this work. 

Cheers,
Authors of this paper.

[Public Comment · li bofang · 14 Jan 2017]
**GloVe model**

Dear reviewers,

We have analyzed GloVe model and updated our paper:
   -The overall tendency of GloVe with different contexts is similar to Skip-Gram. 
   -GloVe is more sensitive to different contexts than Skip-Gram and CBOW, which is probably due to its explicitly defined/optimized objective function.
Please see our revised paper for more details.

The source code at Github is also updated to support GloVe. (

[Final Decision · Program Chairs · 06 Feb 2017]
**ICLR committee final decision**

Reviewers agree that the findings are not clear enough to be of interest, though the effort to do a controlled study is appreciated.